# A Preliminary Direct Comparison of the Inflammatory Reduction and Growth Factor Production Capabilities of Three Commercially Available Wound Products: Collagen Sheet, Manuka Honey Sheet, and a Novel Bioengineered Collagen Derivative + Manuka Honey + Hydroxyapatite Sheet

**DOI:** 10.3390/ijms231810670

**Published:** 2022-09-14

**Authors:** Isaac Rodriguez, Tricia Conti, Nina Bionda

**Affiliations:** 1SweetBio, Inc., Memphis, TN 38111, USA; 2iFyber, LLC, Ithaca, NY 14850, USA

**Keywords:** wound healing, collagen, manuka honey, tissue engineering, biomaterials, inflammation, cytokines

## Abstract

Many commercially available wound products focus on improving one stage of the wound healing cascade. While this targeted approach works for specific wounds, there is a need for products that can reliably and comprehensively progress a wound through multiple stages. This preliminary in vitro study was performed to directly compare the inflammatory reduction and growth factor production effects of three commercially available wound care products: a collagen sheet (COL), a Manuka Honey Calcium Alginate sheet (MH), and a novel bioengineered sheet comprised of a collagen derivative (gelatin), Manuka honey, and hydroxyapatite (BCMH). Macrophages and human dermal fibroblasts were directly seeded on all three commercial products, and supernatants were analyzed for inflammatory markers and growth factors, respectively. Comparing the MMP-9/TIMP-1 ratio, BCMH resulted in 11× lower levels of this inflammation biomarker compared to COL, and 3× lower levels compared to MH. Both the COL and BCMH products created an environment conducive to expression and release of relevant growth factors, while the MH product showed the lowest levels of growth factor expression of all three commercially available products tested. The favorable 11× lower MMP-9/TIMP-1 ratio observed with the BCMH product compared to the COL product suggests that the BCMH products provided a superior comprehensive approach to healthy progression of the wounds by providing an additional benefit of reducing the inflammatory response in vitro.

## 1. Introduction

Regardless of the type of wound, the healing process must cascade through four stages: hemostasis, inflammation, proliferation, and remodeling [1]. Real-time, bedside analysis for identifying exactly where the wound is in this cascade is not yet a practical standard of care. Compounded with difficult patient scenarios (non-compliance, comorbidities, economic disparity, etc.), the need for a more comprehensive approach to address the wound across the cascade is critical to progress healing efficiently and effectively. This is especially vital in chronic wounds that impact millions of patients and cost tens of billions of dollars to the U.S. healthcare system [2].

A wound can become chronic (non-healing) if this cascade is disrupted [3]. Many wounds become chronic by being stalled in the inflammatory phase due to detrimental levels of inflammatory markers. Specifically, matrix metalloproteinases (MMPs) and tissue inhibitors of MMPs (TIMPs) are important inflammatory markers as they play a major role in the degradation of the extracellular matrix and also must be present in the right amounts at the right times [4,5,6]. Specifically, higher levels of MMPs and lower levels of TIMPs are known to be top offenders in the prolonged inflammation state of the wounds, ultimately resulting in chronic wounds [4,6,7]. This has been proven clinically where fluid from chronic wounds was collected and analyzed. The results demonstrated that these chronic wounds had an increased MMP-9/TIMP-1 ratio, which was correlated with poor wound healing [5,8,9,10,11].

Once a wound progresses through the inflammatory phase, growth factors then play a key role in signaling cell proliferation, migration, and differentiation. In chronic wound microenvironments, levels of growth factors have been shown to be in lower amounts, which can further disrupt healing [12,13]. Therefore, new products that can control the microenvironment by decreasing the MMP-9/TIMP-1 ratio while promoting the secretion of pro-healing growth factors will create a new generation of products that provide this more reliable and comprehensive approach, an approach that gives more patients in difficult scenarios with complicated wounds a better chance to heal.

Considering the wound care products that have the potential to address some of these aspects, two materials of interest are collagen and Manuka honey. Collagen and its derivatives are attractive materials for wound products because they act as a sacrificial substrate to buffer high levels of MMPs, which supports progression through the inflammation stage [14]. Manuka honey is also an attractive material as it can kill bacteria, de-regulate MMPs, and upregulate growth factor secretion [15,16]. While these are important benefits in wound care, Manuka honey in its currently available forms has limitations: (1) at higher concentrations it can be cytotoxic, which can offset the healing benefits [15,17]; (2) products comprised of a majority of honey are sticky gels, pastes, or sheets, making them difficult to handle; and (3) current products are considered short-term, topical treatments requiring cumbersome, frequent reapplications. Therefore, right sizing the amount of Manuka honey exposed to the wound microenvironment and providing it in a more useable, long-lasting form is critical to take advantage of the benefits that Manuka honey provides. A novel bioengineered sheet (APIS^®^, SweetBio, Inc., Memphis, TN, USA) comprised of a collagen derivative (gelatin), Manuka honey, and hydroxyapatite is now available in the U.S. Given its synthesized composition, it has the potential to overcome the above-mentioned shortcomings of the individual collagen and Manuka honey materials.

In order to assess the potential of a comprehensive approach to treat complex wounds, this preliminary in vitro study was performed to directly compare the inflammatory reduction and growth factor production effects of three commercially available wound care products: a 100% collagen sheet, a 95% Manuka honey sheet, and the novel bioengineered sheet comprised of a collagen derivative (gelatin), Manuka honey, and hydroxyapatite.

## 2. Results

### 2.1. MMP Array Assay

The results of the MMP array assay are shown in Figure 1. Each fluorescent signal was normalized based on the reference array, which was designated as the control array (i.e., the array incubated with sample buffer only) from the slide on which the standards were run. After normalization, standard curves were plotted using the data from the arrays incubated with protein standards, and the best fit equation for each was used to calculate the amount of secreted protein detected for each cell culture sample (Figure 1).

Proteins detected in the cell culture samples were MMP-1, MMP-9, TIMP-1, and TIMP-2. When comparing the products, the COL and BCMH samples both had similar amounts of TIMP-1 and TIMP-2; however, the amount of MMP-9 was significantly lower for the BCMH samples. As the MMP-1 data exhibited larger standard deviations, it cannot be said with certainty if there were any differences. In contrast to the other dressings, MH samples had low or undetectable levels of all proteins.

### 2.2. Growth Factor Array Assay

The growth factor array results are depicted in Figure 2 and Figure 3. The data was analyzed in the same manner as described for the MMP assay. Note that the fibroblast growth medium contains added growth factors, namely bFGF, EGF, insulin, and TGF beta, so the amount present in the medium was subtracted from the final concentrations obtained from the data analysis. Figure 2 shows the full data plot, while Figure 3 shows the plot with a *y*-axis from 0–20 ng to better visualize the lower range of the data.

As shown in Figure 2, the most abundant proteins were BMP-5 and TGF-beta 1, both of which were present at high levels in the BCMH samples (around 90 and 189 ng, respectively) but low or negligible levels for the other samples. Other proteins detected at relatively high levels were hepatocyte growth factor (HGF) at around 37 ng for the COL samples only and BMP-7 at around 25 ng for the TCP control, BCMH, and the COL samples.

The lower range of the data set is shown in Figure 3. Most notable proteins in this range were FGF-4, which was highest for the TCP control (at about 4.7 ng); GDF-15, which was highest for BCMH samples (10 ng); IGFBP-2, highest for the TCP control at 6.9 ng; and IGFBP-6, which was highest for COL samples at 6 ng. Other proteins detected at appreciable levels were amphiregulin (AR), BMP-4, IGFBP-4, NT-3, and TGF alpha. For these proteins, BCMH samples showed the highest amounts among the treatments for AR, NT-3, and TGF alpha.

## 3. Discussion

### 3.1. MMP Array Assay

When comparing the amounts of protein detected, the most abundant was TIMP-1, which was around 55 ng for the TCP control and 40 ng for both COL and BCMH. TIMP-1 was negligeable in MH samples. TIMP-2 showed a similar trend, albeit at lower amounts than TIMP-1. The TCP control had around 21 ng TIMP-2, COL and BCMH both had around 9 ng, and MH did not have a detectable amount. MMP-1 was detected at around 9 ng for BCMH samples, while COL had around 4 ng and MH around 2 ng. The TCP control samples, on the other hand, did not have detectable levels of MMP-1. Lastly, MMP-9 was present at about 5 ng in the TCP control samples and 3 ng for the COL samples. Both BCMH and MH samples did not show appreciable levels of MMP-9. With the exception of MMP-1, the trends observed for the TCP control and BCMH samples showed agreement with previously published results [18].

Of particular interest was the MMP-9 to TIMP-1 ratio, as increased ratios have been associated with chronic wounds and poor wound healing [5,8,9,10,11]. Table 1 summarizes the amounts of each protein and the corresponding ratio for each treatment. The lowest ratio that was observed was for the BCMH samples, which was around 0.006. This was around 16× lower than the TCP control, 11× lower than COL, and 3× lower than MH. This suggests that of the commercially available wound products tested, the BCMH sheet shows the most promise in diminishing the MMP-9/TIMP-1 ratio and progressing healing of chronic wounds.

### 3.2. Growth Factor Array Assay

As mentioned in the results section, the most abundant proteins were BMP-5 and TGF-beta 1, both of which were present at high levels in BCMH samples but at low or negligible levels in the other commercially available wound products. TGF-beta 1 plays a role in several cellular functions, including the regulation of growth and proliferation, differentiation, cell motility, and apoptosis [19]. BMP-5 is part of the TGF-beta superfamily and is involved in many developmental processes, such as cartilage and bone formation and neurogenesis [20]. Furthermore, BMPs in general are implicated in a number of processes related to wound healing, including keratinocyte migration and proliferation, though their roles have not been completely elucidated and likely differ depending on the stage in the wound healing process [21].

Overall, MH samples showed the lowest levels of growth factor expression of all three commercially available products tested. It is possible that the high concentration of Manuka honey within the MH product was cytotoxic to the cells, a known characteristic of Manuka honey [15,17]. Between the COL and BCMH products, there is a notable difference in the expression profile of the growth factors evaluated in this study, both positive. Collectively, the data indicate that both the COL and BCMH products create an environment conducive to expression and release of relevant growth factors.

## 4. Materials and Methods

### 4.1. Materials

Three wound care products commercially available in the U.S. were used in this study: a Bovine 100% Native Collagen sheet (COL), a Manuka Honey Calcium Alginate sheet (MH) containing 95% Active Leptospermum Honey, and a novel bioengineered sheet comprised of a collagen derivative (gelatin), Manuka honey, and hydroxyapatite (BCMH).

### 4.2. MMP Array Assay

The secretion of matrix metalloproteinases (MMPs) by human macrophages seeded on three commercially available wound products (COL, MH, and BCMH) was directly compared. The macrophages were derived from human monocytes (THP-1, ATCC TIB-202, Manassas, VA, USA) by differentiation in vitro. Briefly, monocytes were cultured in medium containing 25 nM phorbol 12-myristate 13-acetate (PMA) for 48 h, followed by a 24-h rest period in culture medium without PMA. The cells were then prepared for seeding by washing once with sterile 1X PBS and detaching from the tissue culture flask using a 0.05% trypsin/EDTA solution.

The assay was performed in a 12-well tissue culture plate with the following treatments assessed in triplicate: COL, MH, BCMH, and tissue culture plate (TCP) control. To facilitate interaction of the cells with the dressings, an anti-adherence rinsing solution (Stemcell Technologies, cat. # 07010) was used to pre-treat the culture plate. Each well was rinsed with 1 mL of the anti-adherence solution, followed by a rinse with 1 mL of sterile 1X PBS. Wells for the TCP control were left untreated to serve as a cell culture growth control.

The test materials were prepared by using a sterilized 16 mm diameter hollow steel punch to aseptically cut circular coupons of each material. COL and MH samples were transferred to the assay plate immediately after cutting, while BCMH samples were first hydrated for 2 min in sterile 0.9% saline, according to the instructions for use. The materials were then immediately seeded with macrophages at a concentration of 2.7 × 105 cells/well (2 mL of cell suspension per well). The plate was incubated in a humidified incubator at 37 °C with 5% CO_2_ for 24 h, after which the cell culture medium was collected from each well, centrifuged to remove any cells or particulates, and aliquoted and stored at −20 °C.

The Quantibody^®^ Human MMP Array 1 assay was performed according to the manufacturer’s protocol. The slides were removed from the packaging and allowed to dry for about 3 h at room temperature prior to use. The wells were then blocked using 100 µL of the sample diluent for 30 min at room temperature with gentle rocking. The assay protein standards were prepared by reconstituting the lyophilized protein mix and then performing a series of six 1:3 dilutions. Following the blocking period, the buffer was removed, and 100 µL of either the samples or the protein standards were added to each well. To minimize background, the cell culture samples were diluted 1:2 with sample diluent prior to being added to the slides. The slides were incubated with the samples and standards overnight at 4 °C on a rocking platform. The next morning, the slides were allowed to warm to room temperature for 1 h before proceeding to the next step. The samples were aspirated, and the slides were washed five times with 150 µL wash buffer I and two times with 150 µL wash buffer II. Each wash was incubated for 5 min at room temperature on the rocker. After washing, the biotinylated detection antibody was added to each well at an amount of 80 µL and the slides were rocked at room temperature for 2 h. The slides were then washed as previously described. Next, 80 µL of the Cy3 equivalent dye-conjugated streptavidin was added to each well and the slides were incubated on the rocker at room temperature for 1 h, protected from light. The slides were then washed five times with wash buffer I, as previously described. After washing, the gasket was removed from each slide, and the slides were placed in a 4-slide holder/centrifuge tube. The tube was filled with 30 mL of wash buffer I and rocked at room temperature for 15 min. Wash buffer I was then decanted, 30 mL of wash buffer II was added, and the tube was rocked at room temperature for 5 min. Wash buffer II was decanted, and the slides were removed and rinsed gently with DI H_2_O to remove any residue. Gentle aspiration was used to remove all remaining water droplets from the slides, and the dried slides were then stored at 4 °C until being shipped to RayBiotech for scanning and data extraction. Data analysis was performed using RayBiotech’s free data analysis tool.

### 4.3. Growth Factor Array Assay

Growth factor production by primary human dermal fibroblasts (adult (HDFa), ATCC PCS-201-012, Manassas, VA, USA) seeded on three commercially available wound products (COL, MH, and BCMH) was directly compared. The assay was performed in the same manner as the macrophage assay—a 12-well plate was pretreated with anti-adherence rinsing solution, then 16 mm diameter circular coupons of each material were added and seeded with 2.5 × 105 fibroblasts in a 2 mL suspension. Following a 24 h incubation period at 37 °C with 5% CO_2_, the culture medium from each well was removed, centrifuged, then aliquoted and stored at −20 °C. The procedure for performing the Quantibody^®^ Human Growth Factor Array Q1 assay was the same as described in the MMP Array assay section.

### 4.4. Statistical Analysis

For each MMP or growth factor, JMP software was used for ANOVA (analysis of variance) using values from three sample replicates with three assay replicates each (n = nine values for each of the four dressing types). Alpha (α) for the analysis was 0.05. Tukey–Kramer analysis was then used to compare all pairs of dressings for significant differences, where significantly different corresponded to *p* < 0.05.

## 5. Conclusions

This preliminary study was performed to compare the biological effects elicited by the three commercially available wound products: a Bovine 100% Native Collagen sheet (COL), a Manuka Honey Calcium Alginate sheet (MH) containing 95% Active Leptospermum Honey, and a novel bioengineered sheet comprised of a collagen derivative (gelatin), Manuka honey, and hydroxyapatite (BCMH).

In particular, the effects on secretion of MMPs and TIMPs by human macrophages as well as the production of growth factors by human dermal fibroblasts after a 24 h incubation period were evaluated. The lower MMP-9/TIMP-1 ratio results suggest that the BCMH product may provide a greater benefit in promoting the healing of such wounds than the other products that were tested. This result is even more remarkable in the context of the role collagen plays in the in vivo wound healing process given that the COL product is comprised of 100% collagen. On the other hand, MH samples had low or undetectable levels of all biomarkers in this assay. Results from the growth factor array assay indicated the production of a variety of growth factors to varying degrees in response to incubation with the test materials. COL and BCMH samples showed different profiles of growth factor expression, but both were favorable. MH again did not appear to promote expression of any of the biomarkers evaluated in this study. In fact, compared to the tissue culture levels, it decreased expression of some of the growth factors, which could be explained by the cytotoxic nature of Manuka honey at high concentrations.

Inflammatory markers and growth factor secretion were chosen as endpoints in this study as they represent the early and later stages of the wound healing cascade, respectively. Many commercially available wound care products focus on improving one stage of the wound healing cascade. While this targeted approach works well for specific wounds, there is a need for products that can reliably and comprehensively progress a wound through multiple stages. This comprehensive product approach would allow for flexibility during treatment by not having to predict which single product will progress the wound based on the likely stage the wound is in. Wounds are complex, and comprehensive products should exist to simplify their treatment.

Cumulatively, the results from this preliminary study indicated that the COL and BCMH products provided a comprehensive approach (with respect to controlling inflammation and growth factor release) to progressing wound healing, unlike the MH product. Furthermore, the critical MMP-9/TIMP-1 inflammatory marker ratio for BCMH products was 11× lower compared to COL products. This favorable low inflammatory marker ratio promotes a microenvironment more conducive to wound healing, which suggests that the BCMH products provided a superior comprehensive approach to wound healing.

It is acknowledged that the in vitro results in this preliminary study do not represent a direct clinical correlation. However, the BCMH sheet has a published clinical evaluation demonstrating a 4.1-week closure time of lower extremity chronic wounds in nine patients [22]. This favorable wound closure rate could be explained by the results of this study, where the BCHM sheet can reduce inflammatory markers and increase endogenous growth factor secretion in vitro. Future studies can be performed to further explore the BCMH impact on the levels of these markers in the clinical wound microenvironment and their association with wound closure rates. Future in vitro studies of interest include proinflammatory cytokine and chemokine production by activated macrophages, level of oxidative stress, antimicrobial efficacy, 3D imaging of cells on the dressing surfaces, and skin cell viability, proliferation, and migration.

## Figures and Tables

**Figure 1 ijms-23-10670-f001:**
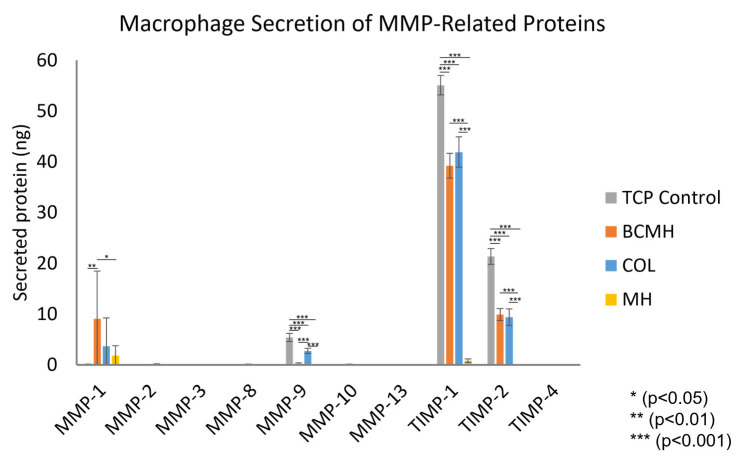
Amounts of MMPs secreted by human macrophages cultured 24 h with each test material (*n* = 9).

**Figure 2 ijms-23-10670-f002:**
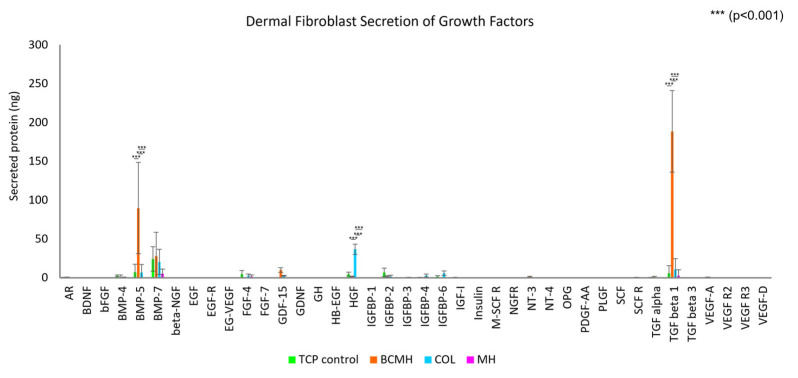
Amounts of growth factors secreted by primary dermal fibroblasts (5 × 105 cells per well) cultured 24 h with the test materials. *n* = 9 for each material.

**Figure 3 ijms-23-10670-f003:**
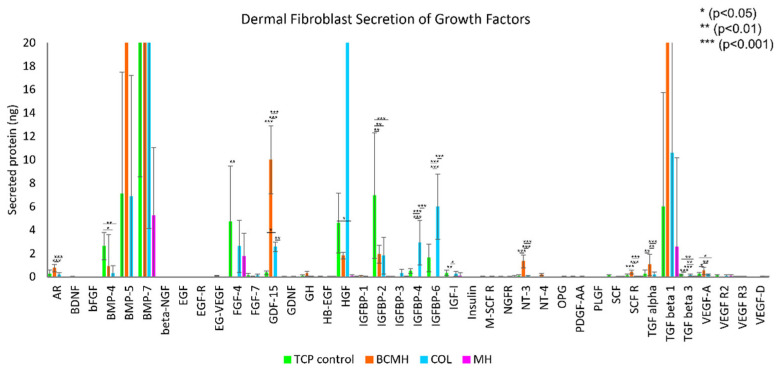
Growth factor data from 0–20 ng, to better visualize lower range of data set. *n* = 9 for each material.

**Table 1 ijms-23-10670-t001:** MMP-9 and TIMP-1 levels and corresponding ratios.

Treatment	MMP-9 (ng)	TIMP-1 (ng)	MMP-9/TIMP-1 Ratio
TCP Control	5.38	55.08	0.098
BCMH	0.23	39.24	0.006
COL	2.74	41.92	0.065
MH	0.02	0.82	0.018

## Data Availability

Data available upon request. Contact isaac@sweetbio.com.

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
