# Peer review of "A Preliminary Direct Comparison of the Inflammatory Reduction and Growth Factor Production Capabilities of Three Commercially Available Wound Products: Collagen Sheet, Manuka Honey Sheet, and a Novel Bioengineered Collagen Derivative + Manuka Honey + Hydroxyapatite Sheet"

_ijms, 2022, doi:10.3390/ijms231810670_

Round 1

Reviewer 1 Report

The authors conducted an in vitro study to compare the anti-inflammatory activity and growth factor production of 3 commercially available wound dressings. The authors intended to identify a multifaceted dressing which could promote wound healing.

Comments:

1.      Described how the statistical analysis was performed in this study. Also, state the sample size for every assays.

2.      The authors need to mention the sample size in every figure legend. The authors should also mention if any significant differences were detected between the groups.

3.      The authors selected the MMPs and TIMPs as the inflammatory markers in this study. Ideally, the authors should measure the levels of proinflammatory cytokines and chemokines produced by the activated macrophages.

4.      One of the factors that lead to the formation of chronic wounds is excessive ROS production or aberrant ROS removal that resulted in elevated level of oxidative stress. As Manuka honey has high antioxidant activities, the authors should study the level of oxidative stress of each dressing.

5.      The authors should study if the dressings can promote skin cell proliferation and migration which are essential in wound healing.

Reviewer 2 Report

Dear authors.

The subject matter is very relevant to the problems of modern medicine. Wound healing is still an important medical problem and problems in wound healing are increasingly linked to civilisation problems.

The meta-studies presented in the study are very interesting with regard to the application in wound healing, however, the work presented here is not a complete study.

(1) The research presented is only the beginning of the research process. The research results presented should be complemented by a set of in vitro studies covering the effects on proliferation, migration, viability, oxidative stress etc. of the findings. 

(2). the paper lacks statistical analyses and calculations.

(3) The research results presented should be regarded as preliminary. They can be published as a short report. 

(4) The work also needs to be supplemented with photographic documentation of cell cultures.

Dear authors, I encourage you to do further research and re-present the results of your work. 

Round 2

Reviewer 2 Report

Dear Authors. Thank you for your replies. I accept the publication as presented.